# Colloidal Lignin Particles as Adhesives for Soft Materials

**DOI:** 10.3390/nano8121001

**Published:** 2018-12-03

**Authors:** Maija-Liisa Mattinen, Guillaume Riviere, Alexander Henn, Robertus Wahyu N. Nugroho, Timo Leskinen, Outi Nivala, Juan José Valle-Delgado, Mauri A. Kostiainen, Monika Österberg

**Affiliations:** 1Bioproduct Chemistry, Department of Bioproducts and Biosystems, School of Chemical Engineering, Aalto University, P.O. Box 16300, FI-00076 Aalto, Espoo, Finland; guillaume.riviere@aalto.fi (G.R.); karl.henn@aalto.fi (A.H.); robertus.nugroho@aalto.fi (R.W.N.N.); timo.leskinen@aalto.fi (T.L.); juanjose.valledelgado@aalto.fi (J.J.V.-D.); monika.osterberg@aalto.fi (M.Ö.); 2VTT Technical Research Centre of Finland Ltd., P.O. Box 1000, FI-02044 VTT Espoo, Finland; outi.nivala@helsinki.fi; 3Biohybrid Materials, Department of Bioproducts and Biosystems, School of Chemical Engineering, Aalto University, P.O. Box 16100, FI-00076 Aalto, Espoo, Finland; mauri.kostiainen@aalto.fi

**Keywords:** lignin, nanoparticle, protein, nanocellulose, fibril, enzyme, heat, self-assembly, cross-link

## Abstract

Lignin has interesting functionalities to be exploited in adhesives for medicine, foods and textiles. Nanoparticles (NPs) < 100 nm coated with poly (_L_-lysine), PL and poly(_L_-glutamic acid) PGA were prepared from the laccase treated lignin to coat nanocellulose fibrils (CNF) with heat. NPs ca. 300 nm were prepared, β-casein coated and cross-linked with transglutaminase (Tgase) to agglutinate chamois. Size exclusion chromatography (SEC) and Fourier-transform infrared (FTIR) spectroscopy were used to characterize polymerized lignin, while zeta potential and dynamic light scattering (DLS) to ensure coating of colloidal lignin particles (CLPs). Protein adsorption on lignin was studied by quartz crystal microbalance (QCM). Atomic force microscopy (AFM) was exploited to examine interactions between different polymers and to image NPs with transmission electron microscopy (TEM). Tensile testing showed, when using CLPs for the adhesion, the stress improved ca. 10 and strain ca. 6 times compared to unmodified Kraft. For the β-casein NPs, the values were 20 and 8, respectively, and for the β-casein coated CLPs between these two cases. When NPs were dispersed in adhesive formulation, the increased Young’s moduli confirmed significant improvement in the stiffness of the joints over the adhesive alone. Exploitation of lignin in nanoparticulate morphology is a potential method to prepare bionanomaterials for advanced applications.

## 1. Introduction

Technologies focusing on the preparation of adhesives utilizing natural polymers such as proteins and cellulosics for medical, textile and food applications are emerging research fields [1,2,3] However, biorefinery industry produces also lignin by-product, which is still underutilized, even though this aromatic, antioxidative and antimicrobial biopolymer could be an interesting raw material for many value-added applications [4].

Nanocellulose can be produced by several methods [5,6,7,8]. It is a lightweight, transparent and biodegradable polymer. Hence, nanocellulose fibrils (CNF) as well as bacterial nanocellulose (BC) are excellent raw materials for tissue regeneration and replacement [9,10,11,12]. Major challenges for the exploitation of CNF include the ability to disperse colloidal material with different formulations. Surface functionalization of the fibrils could be an attractive method to improve stability, functionality and compatibility of the nanomaterial with selected matrices. For example, poly (_L_-lysine, PL) coated wax particles assembled on CNF surface yield hydrophobic fibrils [13]. Furthermore, capability to obtain tight bonding between tissue edges to prevent bleeding with excellent gas barrier properties and to achieve strong mechanical strength for the sealant, are crucial properties for the medical adhesives to support wound healing and tissue deformation during the recovery [14,15,16].

Due to excellent solubility and biocompatibility, regenerated silk proteins have been used in medical applications such as textiles, implants and materials for controlled drug release. Deposition of silk fibrin on polymeric surfaces is a remarkable challenge [2]. Enzymatic cross-linking with transglutaminase (Tgase, EC 2.3.1.13) catalyzing cross-links between glutamines and lysines has been used to stabilize proteins against chemicals and proteases [17]. Furthermore, Tgase has been used to graft silk proteins onto damaged wool fibers to improve strength of the surfaces and to degrease felting shrinkage during washing [18].

Foods such as edible coatings are nearby medical applications. Food packages based on petroleum-based raw materials are not biodegradable. They have poor oxygen barrier properties possibly leaching harmful compounds into foods. Water-soluble edible coatings based on dairy proteins could be excellent alternatives for these packages. Protein coatings have good gas barrier properties and no bad flavor or taste [19,20,21]. For example, nanospheres prepared from caseins form opaque films and could be used to coat foods as well as biological tissues [22]. However, poor water resistance and mechanical strength of the casein coatings needs to be improved to meet full applicability of the nanomaterial in above applications [19].

Silkworm adhesive is an excellent biomimetic model for the preparation coatings for nanoparticulate bonding agents [23] since many technologies for tissue engineering and surgery rely on nanoparticle (NP) based adhesion [24]. Strong and rapid adhesion between hydrogels is feasible at room temperature by spraying hard NPs on the surfaces and bringing them into contact. Tight adhesion between the soft materials is based on the NPs’ ability to adsorb tightly onto surfaces, where they act as connectors between polymer chains dissipating energy under stress [24]. Thus, tailored colloidal lignin particles (CLPs) prepared from technical lignin could be interesting nanomaterials to be used as additives in adhesives and coatings. Different CLPs could be produced in the laboratory and semi-industrial scale [25,26,27,28,29,30,31,32,33,34,35,36]. Enzymatic cross-linking could be an attractive method to increase porosity of NPs in addition to stability improvement against organic solvents [37,38,39]. Including small molecules in the hydrophobic core of CLPs antioxidant and antimicrobial property of the particles could be enhanced [40,41]. Specificity of the particles could be tailored via surface modification [34,42,43,44].

In this contribution, tiny CLPs including bilayer polypeptide modifications were prepared from Kraft using self-assembling to tailor CNF surfaces with heat treatment. Furthermore, larger protein coated CLPs were prepared and enzymatically cross-linked for adhering skin tissue (chamois). Finally, water-soluble adhesive formulation was used to demonstrate effect of various NPs for the adhesion of soft chamois specimens. It was concluded that nanoparticle architecture could be an interesting general platform for the preparation technical lignin-based nanobiomaterials for advanced applications.

## 2. Materials and Methods

### 2.1. Chemicals

Reagent grade chemicals and solvents for the CLP preparation and modifications were purchased from Sigma-Aldrich (Steinheim, Germany). Water soluble Pritt adhesive (Henkel AG & Co, Düsseldorf, Germany) was purchased from a department store in Finland. Throughout the study, Milli-Q water was used in the aqueous solutions.

### 2.2. Proteins

Mixture of serum proteins (pI 5.2–7.8), casein from bovine milk (mixture of α-, β-, λ- and κ-subunits), gelatin (MW 47 kDa, pI 7.0–9.0), bovine serum albumin (BSA, 66 kDa, pI 4.8–5.6) and purified β-casein (MW 24 kDa, pI 4.6–5.1) were purchased from Sigma-Aldrich (Germany). For analyses, gelatin was dialyzed (cut-off 21 kDa) and freeze-dried. Due to low solubility, β-casein was first dissolved in H_2_O and vortexed at room temperature following dilution with H_2_O (1 mg mL^−1^) and pH adjustment (pH 3.0). After 2 h, the solution was vortexed, ultrasonicated and filtrated. Collagen IV (Col IV) from human placenta (Sigma Aldrich, USA) was treated according to Goffin et al. [45] PL peptide (0.1 m-% in H_2_O *w/v*, MW 150–300 kDa, pI 9.0) and sodium salts of poly(l-glutamic acid, PGA) peptides (MW 50–100 kDa and 15–50 kDa) diluted with water (1 mg mL^−1^) were ordered from Sigma-Aldrich (Germany). Chamois for the adhesion experiments was purchased from Biltema (Espoo, Finland).

### 2.3. Enzymes

Low redox *Melanocarpus albomyces* laccase (MaL, pH-range 5.0–7.5) was overproduced in *Trichoderma reesei*. High redox *Trametes hirsuta* laccase (ThL, pH-range 4.5–5.0) was produced in its native host following chromatographic purification [46,47]. The reactivities of the enzyme preparations were determined against 2.2-azinobis-(3-ethylbenzothiazoline)-6-sulfonate (ABTS) at pH 4.5 in 25 mM Na-succinate buffer [46] using Perkin Elmer Lambda 45 spectrophotometer (USA) at 436 nm (ε = 29.300 M^−1^ cm^−1^). For ThL (3.5 mg mL^−1^), the activity was 5270 nkat mL^−1^ and, for MaL (8.1 mg mL^−1^), it was 2050 nkat mL^−1^. Tgase (pH-range 4–9) [17] was purchased from Activa MP Ajinomoto (Japan). After further purification, the enzyme activity (8764 nkat mL^−1^) was determined as previously described [48].

### 2.4. Nanocellulose

The preparation of CNF exploited in this study was described by Valle-Delgado et al. [49] CNF was produced using mechanical fibrillation of never-dried, bleached Kraft hardwood birch pulps obtained from Finnish pulp mills using a high-pressure fluidizer (Microfluidics M-110Y) from Microfluidics Int. Co. (Westwood, MA, USA). No pre-treatments were used prior to fibrillation. The number of passes through the microfluidizer was 12 and the final dry matter content was 1.35 wt-%. The operating pressure was 2000 bar. The average width of the fibrils was 8–9 nm, length several micrometers and a zeta potential ca. −3 mV. CNF thin films for the CLP coatings were prepared on the silica plates as recently described [49].

### 2.5. Preparation of CLPs

Lignin nanoparticles were prepared from Lignoboost^TM^ purchased from Domtar plant (Plymouth, NC, USA) with minor changes in the procedure [25]. First, lignin (2 g) was dissolved in the mixture of THF and H_2_O (3:1, *v/v*). Then, H_2_O was added in the filtrated solution, filtrated again and finally dialyzed (Spectra/Por**^®^** 1, RC dry dialysis tube, 6–8 kDa) for removal of THF. Concentration of the CLP dispersion was ca. 1.5 mg mL^−1^, average particle size ca. 300 nm and zeta potential ca. −33 mV.

For the preparation of tiny CLPs, lignin was enzymatically oxidized using low and high redox potential laccases. Powdered lignin (1 g) was dissolved in 0.1 M NaOH (700 mL) under constant magnetic stirring at pH 12.5. Then HCl (1 M) was slowly added to adjust the pH 6.0 and 8.0 for the ThL and MaL treatments, respectively. Then, the solution was transferred into a 1 L measuring flask. Due to low reactivity of ThL in alkaline reaction conditions, pH 8.0 was omitted for this enzyme. Then, lignin solutions (330 mL) were oxidized with laccases (500 nkat g^−1^) and magnetically stirred (20 h). The enzymatic reactions were terminated using acid precipitation (1 M HCl). The supernatant (pH 2) was removed using ultracentrifugation (Optima^TM^ L Series, rotor type 70 Ti, Beckman Coulter, Bromma, Sweden) at 6000 rpm (G-force 1000) for 20 min at 25 °C. The precipitate was collected with H_2_O (pH 5.5) and dried (80 °C) prior preparation of tiny CLPs.

After lignin oxidation, the method modified from Lievonen et al. [25] was used to prepare CLPs below 100 nm. Enzymatically treated lignin (0.5–1 mg mL^−1^) and the references (2.1 mg mL^−1^) were solubilized in THF:H_2_O (3:7, *v/v*) and the mixture was stirred (3 h) following filtration with 0.7 µm Whatman GF/F (Sigma-Aldrich, Germany). Then, H_2_O was fast poured into the solution under constant stirring following vigorous mixing (15 min). THF was removed the solution using dialysis (cut-off: 6–8 kDa) under constant flux for 3 days. The aqueous CLP dispersion was filtrated and characterized as previously described [25].

### 2.6. Adsorption of Proteins on Lignin

Adsorption of model proteins on lignin surface was studied by quartz crystal microbalance with dissipation (QCM-D, Q-Sense E4, Sweden) at different pHs [42]. For the analysis, golden plates were oxidized with UV-light (10 min), spin-coated with polystyrene (PS) and lignin (WS 650, Laurell Technologies Corp., North Wales, PA, USA) [50]. PS was dissolved in toluene (0.5 mg mL^−1^) applied twice (50 μL) and dried at 80 °C (30 min). Lignin was coated from the dioxane-H_2_O mixture (85:15 *v/v,* 0.5 mg mL^−1^) and applied four times on a plate and dried as above. Spin-coating sequence was 300 rpm (3 s), 1000 rpm (5 s) and finally 2000 rpm (30 s). Protein samples were dissolved in water (10 μg mL^−1^) at 40 °C and filtrated. For the pH optimization of the β-casein adsorption, it was dissolved in the buffers (50 mM, 0.1 mg mL^−1^): pH 3.0 and 5.0 (citrate), 6.5 (phosphate), 7.4 (PBS) and 8.5 (Tris-HCl). Then, lignin films were stabilized with the buffers (1 h) and exposed to β-casein adsorption (100 µL min^−1^, 25 °C) until stable baseline was detected following rinsing with the buffer (30 min). Masses of the adsorbed β-casein were calculated from the frequencies according to Johannsmann et al. [51]. After β-casein adsorption on lignin film at the optimized pH, the protein coating was cross-linked with Tgase using enzyme dosages 5, 25 and 50 nkat in the measuring cell (40 μL).

### 2.7. Coating CLPs with Proteins

Surfaces of CLPs (1 mg mL^−1^) were coated with β-casein at pH 3.0 using β-casein to CLP mass ratio of 0.00001 to 1. Extent of surface charge modifications and changes in the average particle size of CLPs were analyzed after stabilization of the samples at room temperature overnight. Bilayer protein coated CLPs were prepared by modifying only slightly negatively charged particles first with PL and then with acidic of sodium salts of PGA. In the end of the experiment excess of PGA was added in the solution to ensure maximal coverage of single PL coated CLPs and presence of large amounts of carboxylic acid groups for the esterification reaction, crucial for CNF coating.

### 2.8. Stabilization of Protein Coated CLPs

Surfaces of β-casein coated CLPs were enzymatically stabilized using Tgase. To avoid cross-linking and aggregation of the particles, the enzyme dosage was optimized. In the reactions, Tgase activities varied 5–40 nkat g^−1^. After overnight incubation at room temperature, the enzyme activity was terminated using ultracentrifugation (5000 rpm, 30 min). Supernatant was removed and the precipitate, cross-linked β-casein coated CLPs, were redispersed in H_2_O at pH 3 and pH 7.5 for the stability studies.

### 2.9. Physicochemical Characterization of CLPs

Average particle sizes and zeta potential values of CLP dispersions were analyzed using a Zetasizer (Malvern, Nano-ZS90 instrument, Malvern, UK). The zeta potential values were calculated from the electrophoretic mobility data using Smoluchowski model. Three scans were collected for zeta potential and five scans for the average particle size measurement using dynamic light scattering (DLS) to evaluate the reproducibility of the measurements.

### 2.10. SEC

Polymerization of lignin by laccases was studied by aqueous high-performance gel permeation size exclusion chromatography (HP-GPC/SEC). For the analyses, enzymatically polymerized and cross-linked lignins including molecular weight standards (194 Da to 0.1 kDa) were dissolved in NaOH (0.1 M) in two concentrations (0.1 and 0.5 mg mL^−1^). Weight-average molar mass (MW) of the samples were analyzed by Agilent 1260 Infinity (Agilent Technologies, Espoo, Finland) equipped with a UV detector operating at 280 nm as previously [39,52].

### 2.11. FTIR

Fourier-transform infrared (FTIR) spectra of the lignin samples were recorded using Thermo Nicolet iS50 FTIR spectrometer with iS50 ATR-crystal (Thermo Fisher Scientific, Vantaa, Finland). Analysis of the spectral area (3800–600 cm^−1^) was carried out as duplicate measurements with 32 scans from each sample and averaged prior normalization, which was based on peak area using Excel (Microsoft, Espoo, Finland).

### 2.12. AFM

Atomic force microscopy (AFM) was used to characterize spherical morphology and roughness of CLP surfaces before and after protein coating and enzymatic cross-linking to evaluate aggregation between NPs after the treatments. For the imaging, 10 μL of CLP dispersion was pipetted on a freshly cleaved mica sheet and dried overnight at ambient temperature. All samples were imaged in tapping mode in ambient air using a Multimode 8 AFM equipped with a Nanoscope V controller from Bruker Corporation, Santa Barbara, CA, USA). NCHV-A probes with a fundamental resonance frequency of 320–370 kHz, a nominal spring constant of 40 N m^−1^, and a tip radius below 10 nm were used for imaging. At least three sample areas were imaged from the same mica sheet without further processing of the images except flattening using Nanoscope Analysis 8.15 software from Bruker (USA).

Furthermore, AFM was used to measure adhesion energies between Col IV and lignin, Col IV and gelatin as well as Col IV and casein. For the force measurements the tip less silicon cantilever (CSC38/No Al coating, MicroMasch, Tallinn, Estonia) with a normal spring constant of 40 N m^−1^ was used to study the interactions. Prior to force measurements, the nominal spring constant was determined analyzing the thermal vibration spectra using Sader method [53]. The biomaterial-coated probe was prepared with same method as previously [54]. Adsorption of Col IV onto the colloidal probe was performed in several steps. First, the collagen solution (1 mg mL^−1^) was placed in ice-filled beaker and thawed by sonication (2 × 10 min). Then, the glass probe was surface modified with 5 vol-% 3-aminopropyl triethoxysilane (APTES) dissolved in ethanol to improve physical adsorption of protein on the glass probe (45 min). Unreacted APTES was rinsed with ethanol and dried. The APTES-modified probe was glued with an optical adhesive (Norland Products, Inc., Cranbury, New Jersey, USA) on the free-end of the cantilever with 3D micromanipulator following UV curing (15 min) at the wavelength of 365 nm. After gluing, the colloidal probes modified with APTES were mounted on metallic disc facilitated with double-side tape and few drops of collagen solution were spin-coated (40 s) at 1000 rpm. The collagen-coated probes were dried overnight and rinsed with Milli-Q water before use. The neat glass probe was used a reference.

AFM force measurements were performed using a Multimode 8 AFM NanoScope V controller coupled with a Pico Force (PF) scanner from Bruker (USA) in a liquid mode. The colloidal probe was mounted on the liquid cell and subsequently inserted into the AFM head. Few drops of PBS buffer (pH 7.4) were injected onto sample film and equilibrated (10 min) before the force measurements. The rate of the approach and retraction of the colloidal probe towards the surface was 2 µm s^−1^. At least three random locations were probed to ensure the homogeneity of the film surface. The deflection sensitivity was determined from a freshly cleaved mica surface. The recorded data were converted to the profiles of normalized force and the separation distances, where D = 0 was adjusted to be at the maximum applied load [55]. The measured force profiles were compared to the DLVO theory [56,57]. and the adhesion energy was calculated through the integration over the adhesion area. For the proteins (Col IV), the Hamaker constant for calculation of van der Waals forces was 7.5 × 10^−21^ J [58].

### 2.13. TEM

FEI Tecnai 12 (Hope, CA, USA) operating at 120 kV was used to obtain transmission electron microscopy (TEM) images from the CLP dispersions. For the imaging, 3 µL of the sample was applied on a carbon film supported grid and incubated (2 min). The excess of the solvent was removed by blotting the side of the grid onto paper. Imaging was performed in the brightfield mode with slight under focus.

### 2.14. Sample Preparation

Chamois specimens washed with acetone and dried with filter paper were cut to narrow strips (3.5 cm × 1.0 cm) following stabilization in the standard conditions (25 °C, 50% humidity). The area used for adhesion was 1 cm^−2^. In addition to aqueous NP dispersions (CLP, β-casein and CLP coated with β-casein) in 1 mg mL^−1^ concentration, Tgase (100 nkat cm^−2^) was used for curing β-casein coated CLPs joints. Furthermore, NPs (1 mg mL^−1^) were dispersed in diluted water-soluble adhesive (10 mg mL^−1^) to study the effect of the NPs on the adhesion in the agglutinative formulation. Lignin dissolved in THF (1 mg mL^−1^) and diluted adhesive formulation (H_2_O:THF, 99:1, v:v) in 10 mg mL^−1^ concentration were used as references. After sticking the specimens with NP dispersions (50 µL and 100 µL), the samples where kept under a metal plate (ca. 200 g) in the standard conditions for 3 days prior to tensile testing (MTS400, MTS Systems Corporation, Eden Prairie, MN, USA).

Tiny and bilayer protein coated CLPs (mass ratio 1 g g^−1^ lignin) were linked on the CNF surfaces using esterification reaction between the carboxylic acid groups with hydroxyls of CNF [49,59]. For the analysis, two drops of modified CLP dispersions were coated (4000 rpm, 1 min) on the CNF surface using a spin-coater from Laurell Technologies Corp., (North Wales, PA, USA). Heated up to 105 °C (10 min) following 5 min treatment at 155 °C. To remove unbound particles, CNF surfaces were rinsed with H_2_O and dried under nitrogen flow.

## 3. Results and Discussion

### 3.1. Tailoring CLP Surfaces with Proteins

Proteins adsorb on lignin surface. The extent of the interactions depends on the physicochemical properties of the biomolecule resulting from the three-dimensional (3D) structure and amino acid composition of the protein [42,50]. To show potential to exploit actual by-product from industry, purified β-casein, previously used in wood [60] and food [21] adhesives was used a model protein for the surface functionalization of CLPs.

#### 3.1.1. β-Casein

Adsorption of β-casein on lignin surface was studied at pH 7.4 using QCM-D (Appendix A). Gelatin, serum proteins and PL, commonly used to coat tissues to improve cell adhesion [61], were studied for comparison. Positively charged gelatin (47 kDa, pI 7.0–9.0) at pH 7.4 adsorbed better on lignin surface than smaller negatively charged β-casein (24 kDa, pI 4.6–5.1). Adsorption of serum proteins and polypeptide (PL) were weaker. The increase in dissipation was considerably higher at similar frequency values for the β-casein coating compared to other proteins, indicating that the coating is softer and contains more water (Appendix A).

For the coating of individual CLPs with β-casein, protein adsorption on lignin surface was examined at pH range 3.0–8.5 (Figure 1). It was the highest at pH 3.0 due to positive charge of β-casein in the acidic reaction conditions. A similar adsorbed mass was observed at pH 8.5, but in this case negatively charged β-casein formed particles (ca. 300 nm) that adsorbed on the lignin surface together with the polymeric protein. This is further confirmed when comparing the increase in dissipation (Appendix A). The dissipation is much higher for layers adsorbed at alkaline pH compared to pH 3.5, indicating that these layers are more loosely bound and contain more water due to the nanoparticulate morphology. The formation of β-casein NPs depends on the pH, time, mixing, protein and salt concentration [20].

Hence, pH 3.0 was selected for the coating CLPs with β-casein for further experiments. Figure 2a shows the zeta potential of the CLPs varying from negative (ca. −25 mV) to positive (ca. 25 mV) value when β-casein concentration increased. During the coating, CLPs aggregated when the surface charge of the particles was close to zero (CLP—protein ratio ca. 0.01), as shown in Figure 2b. On the other hand, once clearly positively charged, the protein-coated CLP dispersions were stable for weeks. Large-scale all atom MD simulations [62] have shown that aromatic residues contribute significantly to the protein adsorption on hydrophobic surfaces via strong π–π stacking interactions between *p*2-carbons. The basic residues such as arginine and lysine play equally strong role for the adsorption. The effect of proline residues has been demonstrated recently [42].

In Figure 2 are shown representative TEM images from single CLPs (Figure 2c) including β-casein coated CLPs (Figure 2d) confirming that after protein adsorption, and enzymatic cross-linking (Figure 2e) CLPs remain individual spherical NPs and the aggregation of the particles is minor. During sample preparation on the carbon grid, some of the particles moved close to each other due to water evaporation during drying. The corresponding AFM images are shown in Figure 2f–h and Appendix A. When CLPs were coated with β-casein, very small protein particles could be detected from the background of TEM (Figure 2d) and AFM (Appendix A) images, not visible in the references (Figure 2f,i–k and Appendix A). Apparently, some of these particles adsorbed on CLP surface along with polymeric β-casein since several protruding points (ca. 40 nm) could be imaged from the CLP surfaces by AFM (Appendix A).

#### 3.1.2. Poly(l-glutamic acid)

Feasibility to coat CLP surfaces using selected proteins to maximize specific interactions with the substrate such as CNF surface was evaluated. Thus, negatively charged CLPs were first coated with positively charged PL following modification with PGA containing large number of carboxylic acid groups for the esterification reaction with hydroxyls on nanocellulose surface via fast heat treatment. It was hypothesized that tiny CLPs below 100 nm in size coat single CNF fibrils better than larger particles since the typical width for the nanocellulose fibrils is ca. 5–20 nm and the length several micrometers. The average molecular masses of enzymatically polymerized and cross-linked lignin used for tiny CLP preparation are shown in Figure 3 and the characterization using FTIR spectroscopy in Appendix A. In general, the changes in the FTIR spectra between the references and laccase treated samples were minor due to heterogeneous cross-linking reactions and residual moisture in the samples slightly interfering the interpretation of the spectra.

The appearance of the CLP dispersions at pH 6.0 are shown in Appendix A. The average particle sizes for CLPs prepared from enzymatically oxidized lignin were below 100 nm (Table 1). For the references, the particle size was half of that obtained according to the method of Lievonen at al. [25]. In both cases, the zeta potentials were on the same order of magnitude as previously described [25]. The increased molecular weight, higher hydrophobicity of the polymerized and cross-linked lignin as well as lower concentration enabled tight packing of enzymatically oxidized lignin fast mixing promoting tiny NP CLP formation. In the laccase-catalyzed reactions, the cross-links are formed in lignin via different radical reactions. Enzymatic initiation of the radicalization starts from the phenolic hydroxyl groups of lignin following condensation of the free radicals to covalent chemical bonds [37,38,46,47,63]. The representative AFM images of the different CLPs (Appendix A) verify the spherical and smooth surface structure of the NPs stable for several weeks (Appendix A), as evident from the TEM images (Appendix A). Solid lignin NPs 10–30 nm in size can be produced using mechanical shearing [32] and are also potential modifiers for CNF surfaces.

Bilayer protein coated tiny CLPs (ca. 131 nm) are shown in Appendix A. Significant increase in the average particle size from ca. 200 nm to ca. 400 nm confirmed coating of the NPs with small PGA (15–50 kDa). When using larger PGA (150–300 kDa), the effect on the average particle size and extent of the surface coating was minor. Apparently, shorter polymer chain is more desirable for the modification of CLP surfaces over large ones. However, in the both cases, zeta potential values decreased from ca. 40 mV to 30 mV, verifying dual coating of CLPs. In Appendix A are shown adsorption of PL peptide on the intact CLP surface following adsorption of low and high molecular weight PGA confirming the above conclusions.

In the body, NP-specific protein coronas are formed on hard inorganic NPs in minutes comprising hundreds of proteins. Adsorption of serum proteins on lignin (Appendix A) show that CLP surfaces could be tailored accordingly to increase cellular interaction. Compared to hard NPs, soft CLPs are presumably safer since undesirable penetration of the elastic NPs through the biological membranes is minor. Low cytotoxicity [31,64] and resistance for the enzymatic hydrolysis increase potential to exploit CLPs in value-added applications in medicine and cosmetics [2,20].

### 3.2. Stability of β-Casein Coated CLP

#### 3.2.1. Effect of Enzymatic Cross-Linking

Tailoring CLP surfaces with β-casein allows further modification of the particles with cross-linking enzymes [65]. In vitro studies have shown that Tgase can cross-link proteins in a gel in minutes [17,48]. Hence, to improve stability of β-casein coated CLPs at physiological pHs, for example in stomach (pH 1.0–3.0), duodenum (pH 4.8–8.2) and blood (pH 7.4), Tgase was used to cross-link the β-casein coating. Appendix A shows the increase in the average particle size of CLPs as a function of enzyme dosage. Particles cross-linked with high enzyme dosage above 20 nkat g^−1^ aggregated immediately because covalent bonds were also formed between the individual particles. The zeta potential values (Appendix A) decreased after 24-h incubation and, after seven-day treatment, extensive cross-linking of the NPs was detected by eye. Reasonable enzyme dosage for the cross-linking only β-casein coating was found to be 15 nkat g^−1^. Prior to pH stability studies of the enzymatically cross-linked particles, Tgase activity was removed from the dispersions using ultra-centrifugation (Appendix A). After the treatment, some of the unbound β-casein adsorbed on CLP surfaces increasing the average particle size by ca. 40 nm. Then, the particles were stable for several weeks.

Elasticity of the enzymatically cross-linked β-casein coating was studied by QCM-D (Appendix A). After adsorption of β-casein on lignin film until a stable baseline was observed (Appendix A), Tgase (2 min) was injected into the chamber following cross-linking (10 min) of the protein film. When using the low enzyme dosage (5 nkat), loosely bound β-casein was washed away. However, at the same time, the cross-linking reaction proceeded to a certain extent since a sharp decrease in dissipation was detected due to the formation of elastic networked protein coating. Instead, when using higher enzyme dosages in the measuring cell (25 and 50 nkat), first a small amount of unbound β-casein was removed (tiny increase in the frequency signal). After that, a sharp drop in the frequency was detected due to enzyme adsorption on the β-casein surface. The cross-linking reaction proceeded as in the case of low enzyme dosage. During the enzymatic reaction (10 min), the frequency increased slightly, verifying an accompanied loss of small molecule reaction product (-NH_3_^+^) [66] as well as decreased water binding capacity of the cross-linked protein coating. Tgase injection following stabilization of the cross-linking reaction was repeated three times until stable baseline was observed.

Figure 2c–h shows representative TEM and AFM images of β-casein coated CLPs cross-linked with Tgase. Compared to CLPs coated with β-casein (Appendix A) the surfaces of cross-linked particles were smoother. From the corresponding TEM images (Figure 2d,c), it is evident that also small protein particles in the background were cross-linked to larger particles. Regarding the applicability of the enzymatically stabilized protein coated CLP dispersions, polymerization of free β-casein to nanosized particles [65] improves homogeneity and stability of the dispersion.

#### 3.2.2. Effect of pH

Understanding dispersion properties is essential for the exploitation CLPs in adhesive formulations. CLPs are stable in wide pH range [25]. However, due to better electrostatic stability in alkaline conditions, the reactivity of CLPs is much higher than in acidic conditions [39,63]. For medical, cosmetic and food applications, it is pivotal that tailored CLPs are stable in physiological conditions.

Figure 4a,b shows the average particle sizes and zeta potential values for β-casein coated CLPs at pH 3.0 and 7.4 as a function of time. In acidic dispersion, NPs started to aggregate after four-day incubation, which was evident also from the more positive zeta potential values. After 25 days, protein-coated particles precipitated. At pH 3.0, β-casein was positively charged, but CLP surface was nearly neutral, promoting aggregation of the coated CLPs. At slightly alkaline dispersion (pH 7.4), the stability of β-casein coated CLPs was excellent. The average particle sizes and zeta potential values remained nearly unchanged for 25 days.

To improve stability of the β-casein coated CLPs at pH 3.0, the optimized Tgase dosage (15 nkat g^−1^) was used to cross-link the surfaces of the particles. The stability of cross-linked protein coated CLPs was excellent compared to the non-cross-linked particles (Figure 4a). The average particle sizes remained nearly unchanged for 25 days. Instead, the zeta potential values became slightly more positive due to some instability of CLPs, as explained above. Instead, at pH 7.4 (Figure 4b), the zeta potential values were stable for weeks, but the average particle sizes increased during the first day after enzyme inhibition and solvent exchange. Then, the cross-linked protein coated CLPs were stable for weeks.

### 3.3. Adhesive Interactions

AFM force measurement was used to compare strength of the interactions between lignin and model proteins (Figure 5). Col IV is the main component of the skin and, therefore, understanding the interactions of Col IV with other proteins and lignin could be exploited in wound healing and other biomedical applications.

Long-range attractive interactions were detected when Col IV surface was brought into contact with casein and gelatin (Figure 5a), while no meaningful adhesions were detected when the neat glass surface was used (Figure 5b). The force interactions between Col IV and lignin were three times larger than the ones between Col IV and model proteins (Figure 5c) with adhesion energy being 0.0102 ± 0.0006 nJ m^−1^. Interestingly, the adhesion energy between Col IV and casein as well as Col IV and gelatin was nearly identical at pH 7.4, evident also from the overlapping retraction force profiles (Figure 5a). At high salt concentration, the repulsions detected from the approach force profiles was much longer-ranged than predicted by DLVO theory for pure electrostatic double-layer repulsion (Figure 5d). This reveals that steric repulsions dominate [67]. Based on the approach force profiles separation distance between Col IV and gelatin pair is nearly twofold compared to that of Col IV and casein, while Col IV and lignin was placed in between. Even though casein show lower adhesion energy with Col IV, it is an attractive protein to coat CLPs compared to poorly soluble gelatin having high tendency to aggregate, as recently demonstrated [42]. Thus, purified β-casein containing many reactive sites for enzymatic cross-linking and stabilization is an excellent coat protein for CLPs enabling also fast curing enzymatic means [65].

### 3.4. Applications

Chamois specimens and CNF were used model soft matrixes to demonstrate effect of tailored CLPs to adhere soft materials. Furthermore, enzymatic and chemical heat treatment were exploited for fast covalent curing of the agglutinations.

#### 3.4.1. Agglutination of Chamois Specimens with CLPs

Figure 6a show the chamois leather specimen including CLP dispersion used for adhesion. Tensile testing was used to measure the strength of adhesive joints until break down. The representative stress–strain curves for various NP formulations are shown in Figure 6b,c.

Figure 7a shows the effect of different NPs for adhering protein matrix that was significantly better than that of lignin and commercial water-soluble adhesive. When the number of NPs doubled, the agglutination of the specimens was more than twofold stronger, which was evident from the measured stress values. The differences between the type of the soft material, i.e., lignin or protein, used for the NP preparation was apparent when high NP concentration was used. Adhesive property of β-casein NPs on protein matrix was the highest and for CLPs only half of that. Effect of β-casein coated CLPs on the adhesion was between these two cases showing potential as a method to prepare functional low-cost β-casein NPs from lignin via self-assembly. The tensile strain (Figure 7b) measured from the corresponding samples followed the same order as the stress values. However, the differences between the samples were smaller. When β-casein coated CLPs were used for adhering, the strain values were rather similar between two different quantities. In the case of large number of NPs, the repeatability was poor, presumably due to instability of the β-casein coating at pH 3.0, as explained above. Regarding to the references, lignin powder dissolved in THF and diluted adhesive, the agglutination of the specimens was minor compared to that of NPs. Due to rough, hairy surface of chamois, the NPs spread, adsorbed and penetrated on the soft material better than the thick adhesive formulation, efficiently dissipating energy and retarding fracture of the joints under stress.

Figure 7c,d shows the increased stress and strain values when β-casein coated CLPs were covalently linked to chamois specimens using low Tgase dosage (100 nkat). Increasing the enzyme activity, the strength of the adhesion could be increased [47,62]. These results suggest that protein coated CLPs could also be linked to biological matrixes using enzymes enabling fast curing in moist environments essential for the medical applications. In such seals, CLPs remain single nanoparticles retaining their physicochemical properties since the covalent linkages are formed via protein coating. It is also plausible that cross-links were formed between amine groups in lysine side-chains and the acyl group derived from the carboxylic acid groups present in CLPs due to side reactions of Tgase [17]. If CLPs are coated with tissue specific proteins, the potential rejection reactions could be diminished during the wound healing, yielding small scars important for cosmetic applications. Accordingly, it is presupposed that CLP formulations could be exploited for the preparation of edible coatings for foods.

Furthermore, to study potential to use CLPs as additive in adhesive formulations, NPs and the references were dispersed in the water-soluble adhesive (Figure 7e,f). Unexpectedly, the stress value was the largest (ca. 5%) for the dispersion containing lignin powder compared to that of diluted adhesive. Mixing protein NPs in the adhesive formulation decreased stress value ca. 3%, and for the unmodified CLPs ca. 7%. In the case of tensile strain measurement, the results were opposite. For CLPs, β-casein NPs and β-casein coated CLPs, the tensile strain improved ca. 5%, 12% and 10%, respectively. For lignin powder, the elasticity of the adhesive joint degreased ca. 5% compared to the diluted adhesive.

Figure 8 shows histograms of Young’s modulus for various NPs. The values doubled for the specimens adhered with high number of NPs regardless of the type of the polymer used for the adhesion. Instead, when the NPs were dispersed in the adhesive formulation, the differences between the raw materials became visible. In the case of CLPs, Young’s modulus increased (ca. 70 kPa), but, for the protein-based NPs it degreased (ca. 43 kPa), showing more elastic agglutination due to softer structure of the particles and improved compatibility with the substrate. In the adhesive formulation, the Young’s modulus was the highest for the polymeric lignin showing much stronger adhesive joint, likely due to varying chemical reactions and interactions between the adhesive components and the substrate. Furthermore, interactions between lignin and protein matrix differ from the ones between CLPs and the substrate. In aqueous dispersion, hydrophobic groups are buried inside the CLPs’ hydrophilic groups locating on the surfaces of NPs. Furthermore, the instability of lignin complicated the analyses. The results obtained in the adhesive formulation should be critically considered. Young’s modulus determined for the diluted adhesive was between that of CLPs and lignin powder.

These results show that, when using NP prepared from soft natural matter (technical lignin, proteins and their combinations), it is possible to obtain strong elastic agglutinations between soft surfaces, as evidenced recently [24,68,69]. The type of biopolymer affects significantly the strength, flexibility and elongation of the joint. Studies with soybean-based adhesives containing polymeric lignin pointed out that protein–lignin ratio is the most critical parameter affecting the adhesive interactions [70,71]. In textiles, enzymatic cross-linking has been used to strengthen lignin-containing adhesives [72]. Apparently, exploitation of technical lignin in nanoparticulate morphology in stable dispersion for adhering is an interesting approach for many lignin applications reviewed [73].

#### 3.4.2. Grafting CLPs on CNF Surfaces

Feasibility to coat CNF with varying sized CLPs was also investigated. Figure 9a shows nanocellulose fibrils spin coated with several layers of tiny CLPs (ca. 75 nm). Covalently linked CLPs clearly follow the fibrous morphology of nanocellulose, changing the surface properties of the hydrophilic polymer to that of more hydrophobic lignin surface. Linking of the NPs was obtained via esterification reaction between carboxylic acid groups of lignin and hydroxyl groups of CNF surface via heat treatment [49]. After washing with H_2_O for the removal of unbound CLPs, the fibers accumulated to some extent, however, CLPs remaining linked on the CNF surfaces due to covalent linkages. Hence, it is proposed that CNF modified with CLPs could be excellent bionanomaterials for tissue repairing, having improved stability in body fluids, culture media and resistance against enzymatic hydrolysis.

It was also shown that CNF surfaces (Figure 9b) could be coated accordingly with protein coated CLPs using heat treatment. To increase the reactivity between CNF and protein coated CLPs, nanoparticles were modified first with positively charged PL and then with negatively charged PGA containing large number of carboxylic acid groups. CLPs coated only with PL could not be linked on the slightly negative CNF surface to the same extent. In that case, surface modification of CNF, e.g., via carboxylation to increase negative charges on the surface, is a prerequisite. After the heat treatment, the average particle sizes of the modified CLPs clearly decreased (Appendix A and Figure 9b) due to the formation of covalent linkages between carboxylic acids in PGA and hydroxyls in CLP. After washing with water, most of the bilayer coated CLPs remained on the CNF surface. Feasibility for the bilayer protein coating of CLPs was demonstrated also using QCM-D under identical reaction conditions (Appendix A). PL peptide was first adsorbed to CNF film following adsorption of CLPs yielding intact lignin nanoparticulate surface. Then, PL and PGA (15–50 and 50–100 kDa) were adsorbed on the surface that was finally washed with water. As shown in Appendix A, it was evident that small PGA adsorbed on CLP surface better than large PGA.

Current polymeric tissue adhesives are often unstable in physiological pH requiring complex in vivo analytical systems for the control of the polymerization and cross-linking reactions. Furthermore, they are often toxic [6,69]. Coating fibers with specific CLPs could be an effective way to improve mechanical strength. Adhesives based on nanobridging via hard inorganic NPs [16] as well as colloidal mesoporous silica (CMS) particles [68] have been proposed as alternatives for traditional medical adhesives. Since the adhesion energy is proportional to the surface area of NPs, enzymatically cross-linked CLPs [39] with tailored functionalities could be potential additives for medical adhesives following enzymatic or thermal treatment for fast curing shown above. Additionally, porous structure of CLPs enables quicker decomposition in biological media than inorganic NPs preventing undesirable accumulation in the body. Due to strong autofluorescence of lignin, it is an attractive raw material enabling sensitive real time detection crucial for the development of image-guided procedures for clinical applications.

## 4. Conclusions

Development of green technologies [74,75] for the preparation of bio(nano)materials [76] from the forest process side-streams such as adhesives and coatings [77,78] is increasing constantly. Different CLPs prepared and modified to adhere chamois and to modify CNF surface could be potential additives for various formulations to be exploited for wound sealing, edible coatings and fiber modification for textiles to improve adhesion, hydrophobicity, antimicrobial and antioxidative properties of the coatings. Since the cross-linking methods are fast and feasible in the moist environment, clinical fluorescence imaging of aromatic CLPs is possible. Furthermore, it was concluded that, when using tissue specific proteins, e.g., hydrolyzed from collagen, sericins extracted from silk and caseins fractionated from the dairy side-streams, compatibility of the NPs with the substrates could be enhanced. Compared to NPs prepared solely from proteins, the costs of the raw materials are remarkably lower. Apparently, these results pave the way for the exploitation of technical lignin in multiple forms.

## Figures and Tables

**Figure 1 nanomaterials-08-01001-f001:**
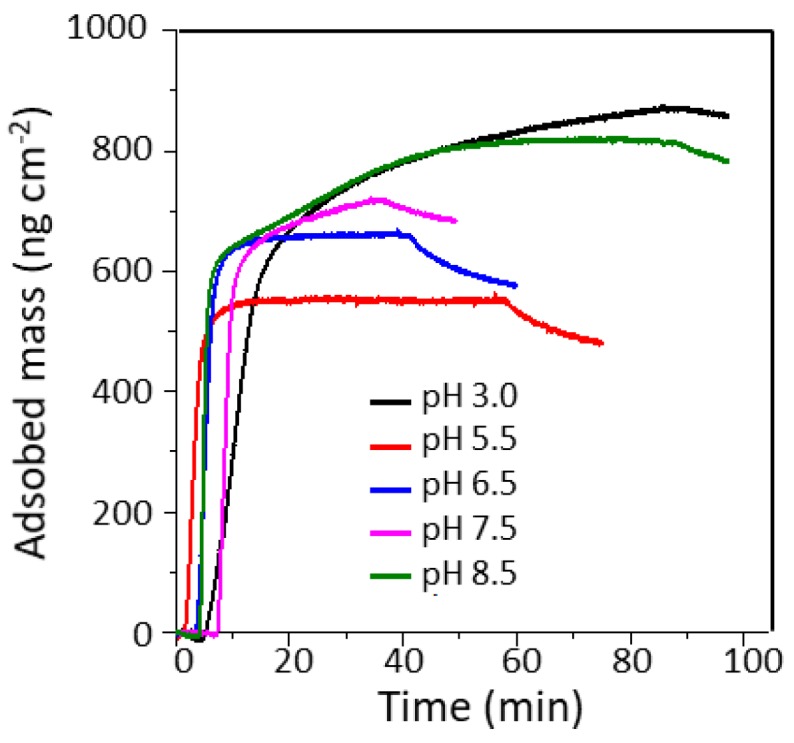
Adsorption of β-casein on lignin thin film at different pH observed using QCM-D.

**Figure 2 nanomaterials-08-01001-f002:**
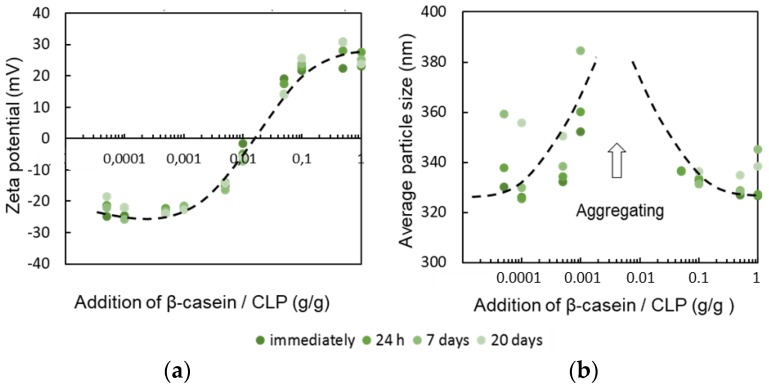
Coating CLPs with β-casein (pH 3.0) evidenced using zeta potential (**a**) and DLC (**b**) measurements as a function of time. TEM images measured from unmodified CLPs (**c**), β-casein coated CLPs (**d**) and β-casein coated CLPs cross-linked with Tgase (**e**). In (**f**,**g**,**h**) are shown representative AFM images from enzymatically stabilized CLPs and in (**i**,**j**,**k**) are presented the corresponding references.

**Figure 3 nanomaterials-08-01001-f003:**
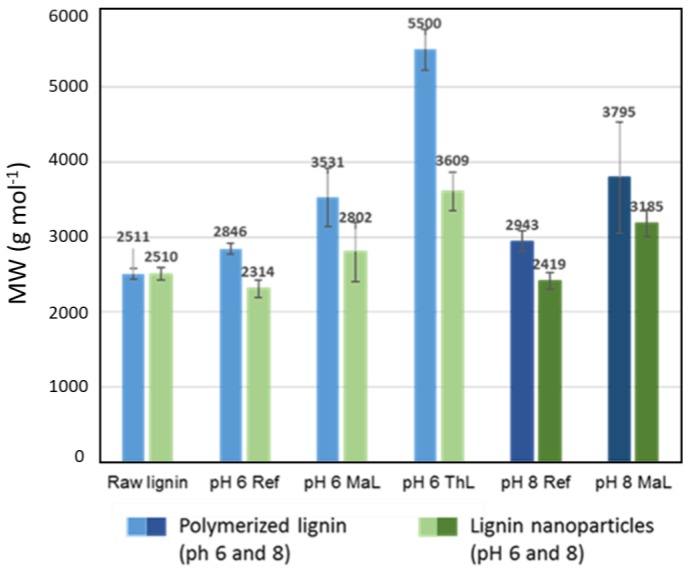
Average molecular masses and standard deviations of the enzymatically oxidized and polymerized lignins. Molecular masses were analyzed also for the CLPs prepared from the enzymatically treated lignins. The smaller values of dissolved CLPs are most likely due to slower solubility of CLPs in alkaline reaction conditions than powdered lignin.

**Figure 4 nanomaterials-08-01001-f004:**
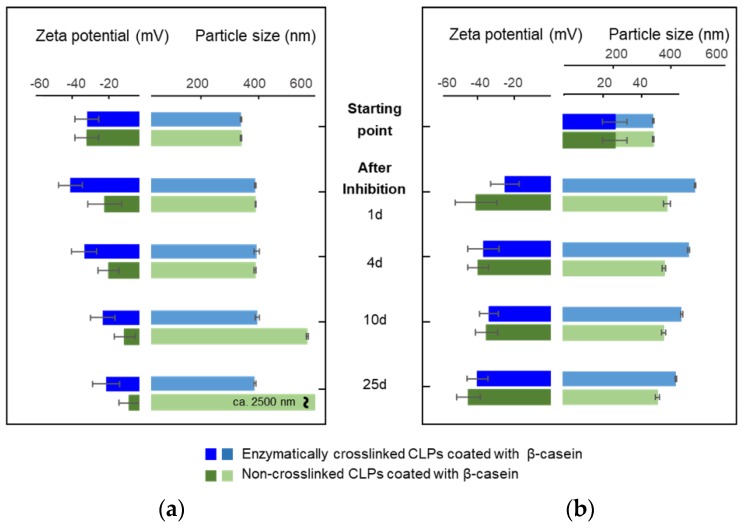
Stability of β-casein coated and enzymatically cross-linked CLPs at (**a**) pH 3.0 and (**b**) pH 7.4 evidenced using average particle size and zeta potential measurements. In both cases, the protein coating following enzymatic cross-linking was performed at 3.0. Enzyme activity was removed using ultracentrifugation following redispersion of the particles at above pH.

**Figure 5 nanomaterials-08-01001-f005:**
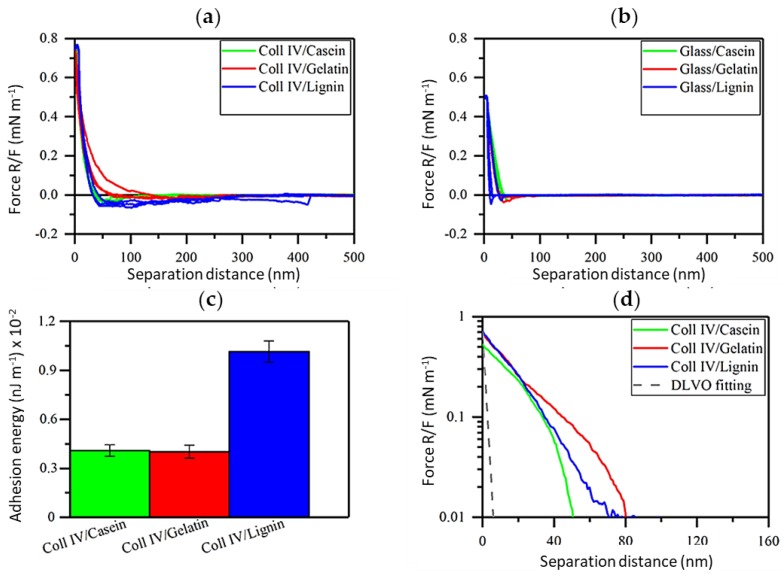
AFM force profiles. Interaction between (**a**) Col IV probe and other proteins and lignin. (**b**) Interaction of the model surfaces with a neat glass probe. (**c**) Corresponding histograms of mean adhesion energy of above interactions including standard deviation. (**d**) DLVO fitting of the data presented in (**b**). The gray dashed line represents the DLVO fitting for PBS (150 mM) at pH 7.4. All force profiles were normalized with the radius of the probe.

**Figure 6 nanomaterials-08-01001-f006:**
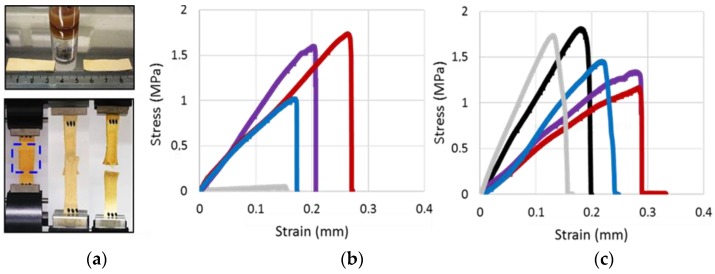
Tensile testing. (**a**) Two chamois specimens to be adhered with CLP dispersion (above). Adhered strips subjected to a controlled tension until failure (below). (**b**) Representative stress–strain curves for the CLPs used for adhesion including the corresponding references. (**c**) Representative stress–strain curves for the same samples as in (b) dispersed in water-soluble adhesive. ● CLPs (blue), ● β-casein NPs (red), ● CLPs coated with β-casein (purple), ● commercial adhesive (black), ● polymeric lignin dissolved in THF (grey).

**Figure 7 nanomaterials-08-01001-f007:**
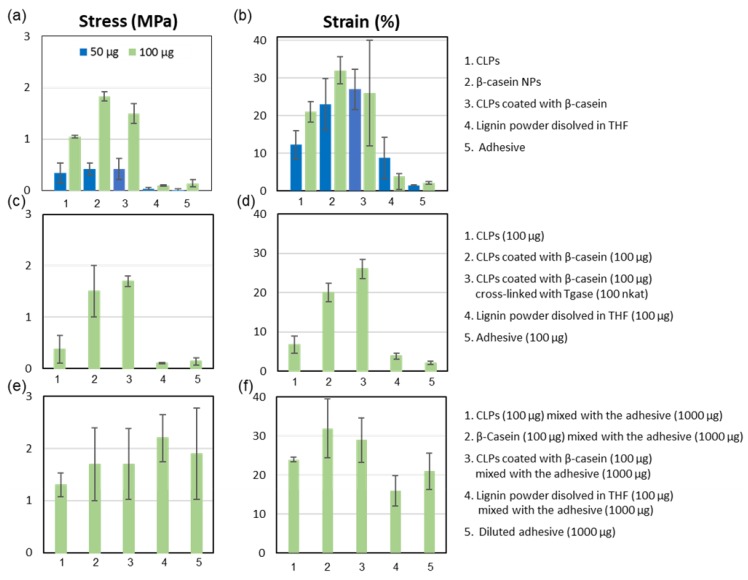
Comparison between tensile stress–strain histograms. (**a**,**b**) Different CLPs including the references at two quantities. (**c**,**d**) Effect of Tgase catalyzed cross-linking on the adhesion. (**e**,**f**) Various NPs including the references dispersed in diluted water-soluble adhesive.

**Figure 8 nanomaterials-08-01001-f008:**
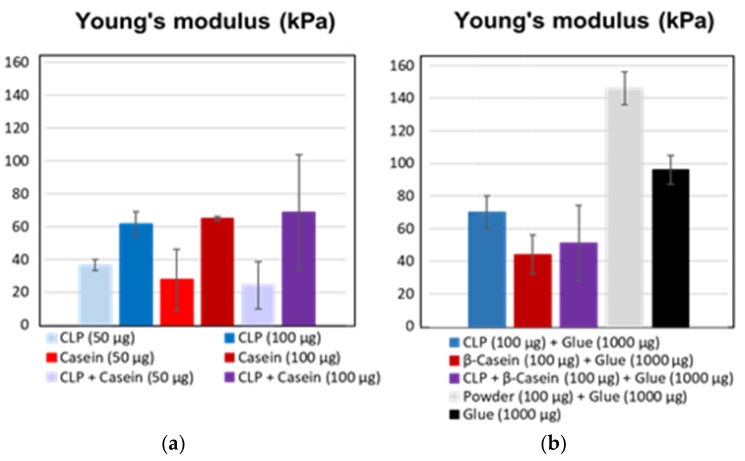
Young’s modulus for (**a**) CLPs, β-casein and CLPs coated with β-casein using two quantities. (**b**) Corresponding histograms for the NPs dispersed in diluted adhesive including the references.

**Figure 9 nanomaterials-08-01001-f009:**
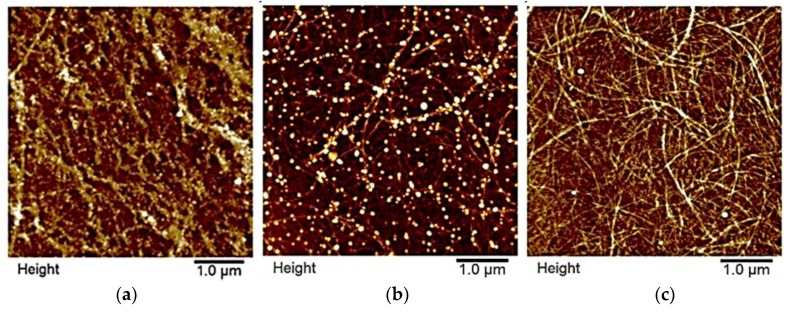
CNF surfaces coated with CLPs: (**a**) Tiny CLPs; (**b**) PL and PGA coated CLPs; and (**c**) reference.

**Table 1 nanomaterials-08-01001-t001:** Characterization of CLPs prepared from laccase-treated lignin.

Sample	Average Size (nm)	Zeta Potential (mV)	PDI
**pH 6.0**			
Reference	131 ± 1	−30 ± 1	0.27
MaL-treatment	75 ± 1	−25 ± 1	0.40
ThL-treatment	82 ± 2	−33 ± 1	0.40
**pH 8.0**			
Reference	125 ± 1	−22 ± 1	0.25
MaL-treatment	65 ± 1	−30 ± 1	0.25
ThL-treatment	-	-	-

(-) Not determined. The enzyme is not reactive at alkaline reaction conditions.

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
