# Peer review of "Colloidal Lignin Particles as Adhesives for Soft Materials"

_nanomaterials, 2018, doi:10.3390/nano8121001_

Round 1

Reviewer 1 Report

The manuscript contains relevant information related to the colloidal lignin nanoparticles for adhesives for soft materials.

The objective of the manuscript in general is clear but treats two different applications with different proteins attached to lignin nanoparticles that makes it sometimes difficult to follow. In addition, there is a lot of methodology mentioned in the results section which should be described in the materials and methods section.

On the other hand, they carry out an in-depth study on the adhesion of the different proteins to the lignin nanoparticles, taking into account different variables and their stabilisation in the case of β-casein.

 However, the authors should clarify, correct or change the following items in the manuscript:

 1.      Can the authors explain what conditions they have used during the use of the high pressure fluidizer? What camera size, pressure, number of steps, etc.

2.      Have the authors analyzed the properties of the resulting CNF?

3.      Was the starting hardwood pulp bleached?  If unbleached, how does residual lignin affect protein adhesion?.

4.      What is the molecular weight of the initial lignin and polymerized lignin resulting from laccase treatments? Didn't NaOH interfere in this analysis?. Because this compound absorbs at 280nm in UV. Is not indicated in the manuscript at the wavelength at which it is measured.

5.      Why does the lignin treatment with laccase facilitate the formation of tiny NPs? Explanation given by the authors (pag 9, line 316) is not very clear.

6. The authors should conclude more clearly the advances found in their research. 

Author Response

Dear Reviewer,

Thank you very much for revising the manuscript.

Your critical comments and feedback improved the

manuscript significantly. Please find enclosed the

updated version.

Kind Regards,

Maija-Liisa Mattinen

Reviewer 2 Report

The manuscript by Mattinen and co-workers details the preparation and application of lignin-based nanoparticles. The work is sufficiently detailed to be reproducible. The topic is timely and of interest to a broad audience. A significant amount of data is presented in the manuscript to support the claims. However, there are several issues that must be addressed prior to publishing.

1. The solvent volumes used (THF, water) under section 2.5 should be reported. Same applies to the other subsections of the methods section.

2. The figures showing the actual data should be provided in high resolution. Currently some of the data are blurred and pixeled, e.g. Figure 1a.

3. Figure 1 should be divided into at least two separate figures, and the panel designations should be placed either at the top or the bottom of the figures but not in a random fashion.

4. The IUPAC-recommended x y^-1 format for units should be consistent throughout the manuscript (e.g. g/mol needs to be corrected in Figure 2 nJ/m, mN/m need to be corrected in Figure 4).

5. The presented work centers on sustainable/green solutions. I short introductory paragraph should be added on the topic with some recent and broad examples to demonstrate the importance and extensive interest in finding sustainable/green solutions (ChemSusChem, 2018, 11, 3640-3648; Green Chem., 2018, 20, 4911-4919; Green Chem., 2017, 19, 3116-3125; Green Chem., 2018, 20, 4901-4910; ACS Catal., 2018, 8, 7430-7438).

6. Closely related works on the use of lignin nanoparticles as adhesives should be acknowledged (Biotechnol Biofuels., 2017, 10, 192.; The Journal of Adhesion, 2018, 94, 814-828; J. Appl. Polym. Sci., 2017, 134, 45124.).

7. What is the accuracy for the PDI measurements? Values are reported down to 0.01 decimal places, which seems too much.

8. The conclusions section should summarize the main research findings, and preferably it should include some quantitative statements. The drawbacks and limitations of the methods should also be mentioned to add a critical edge. On the contrary, the abstract section should contain less quantitative information and should read more general.

Author Response

(The authors gave the same response as above.)

Round 2

Reviewer 1 Report

Thanks to the authors for reviewing the manuscript and taking into account the suggestions of the reviewers. I consider that the manuscript can be published in the present form.

Reviewer 2 Report

The authors have made a thorough revision and the manuscript is ready to be published.